# Synthesis, Characterization, Bioavailability and Antimicrobial Studies of Cefuroxime-Based Organic Salts and Ionic Liquids

**DOI:** 10.3390/pharmaceutics16101291

**Published:** 2024-10-02

**Authors:** Francisco Faísca, Željko Petrovski, Inês Grilo, Sofia A. C. Lima, Miguel M. Santos, Luis C. Branco

**Affiliations:** 1LAQV-REQUIMTE, Department of Chemistry, NOVA School of Science and Technology, NOVA University of Lisbon, 2819-516 Caparica, Portugal; f.faisca@campus.fct.unl.pt (F.F.); z.petrovski@fct.unl.pt (Ž.P.); miguelmsantos@fct.unl.pt (M.M.S.); 2UCIBIO—Applied Molecular Biosciences Unit, Department of Life Sciences, NOVA School of Science and Technology, NOVA University Lisbon, 2819-516 Caparica, Portugal; inesgrilo@fct.unl.pt; 3LAQV, REQUIMTE, ICBAS—School of Medicine and Biomedical Sciences, University of Porto, 4050-313 Porto, Portugal; slima@icbas.up.pt

**Keywords:** antibiotic, antimicrobial activity, cefuroxime, organic salts, ionic liquids, bioavailability

## Abstract

Low oral bioavailability is a common feature in most drugs, including antibiotics, due to low solubility in physiological media and inadequate cell permeability, which may limit their efficacy or restrict their administration in a clinical setting. Cefuroxime is usually administered in its prodrug form, cefuroxime axetil. However, its preparation requires further reaction steps and additional metabolic pathways to be converted into its active form. The combination of Active Pharmaceutical Ingredients (APIs) with biocompatible organic molecules as salts is a viable and documented method to improve the solubility and permeability of a drug. Herein, the preparations of five organic salts of cefuroxime as an anion with enhanced physicochemical characteristics have been reported. These were prepared via buffer-assisted neutralization methodology with pyridinium and imidazolium cations in quantitative yields and presented as solids at room temperature. Cell viability studies on 3T3 cells showed that only the cefuroxime salts combined with longer alkyl chain cations possess higher cytotoxicity than the original drug, and while most salts lost in vitro antibacterial activity against *E. coli, P. aeruginosa* and *B. subtilis*, one compound, [PyC_10_Py][CFX]_2_, retained the activity. Cefuroxime organic salts have a water solubility 8-to-200-times greater than the original drug at 37 °C. The most soluble compounds have a very low octanol-water partition, similar to cefuroxime, while more lipophilic salts partition predominantly to the organic phase.

## 1. Introduction

Cefuroxime is a second-generation broad-spectrum cephalosporin antibiotic, effective against both gram-positive and gram-negative bacteria. First patented in 1971 [1], it was lauded for its potency, attributed to its stability against β-lactamases and diversity, as it was active on strains that previous cephalosporins lacked effectiveness against such as *N. gonorrhoeae*, *N. meningitidis* and ampicillin-resistant strains of *H. influenzae* [2].

Despite its good solubility in water or buffer solvent, the sodium salt of cefuroxime has only been administered intramuscularly or intravenously, as it is not permeable enough to be absorbed through the gastrointestinal tract [3]. Still, a single 500 mg intramuscular dose shows promising pharmacokinetics; in particular, serum concentrations of 8 µg/mL for a period of 2.5 to 3 h exhibit a prolonged half-life of 70 min along with good excretion rates [4]. Yet, administering injectable antibiotics might not be feasible for conditions that do not require hospitalization, and cefuroxime sodium still presented some pharmacokinetic limitations. In the early 90s, cefuroxime axetil was introduced as a solution to this problem [1]. Cefuroxime axetil is the ester prodrug of cefuroxime (chemical structure shown below in Figure 1) that is metabolized into the active drug via nonspecific intestinal esterases [5,6]. The added lipophilicity of the ester moiety makes it readily absorbed in the gut [7]. Peak concentrations of cefuroxime axetil occur in serum approximately 2 h after oral administration; 500 mg tablets achieve concentrations of 4 to 8 µg/mL, which are increased by the presence of food [7].

Cefuroxime axetil is currently the preferred form of the drug for treating a variety of common bacterial infections. The sodium counterpart is usually reserved for more serious infections that require hospitalization, with cefuroxime axetil often used as a step-down drug [8]. However, cefuroxime sodium exhibits no polymorphism, while its acetylation results in distinct crystalline and amorphous forms. Some of these forms may impact chemical stability, pharmacokinetics or interactions with excipients [9]. This modification is one of several techniques to improve solubility and permeability in pharmaceuticals formulations [10]. These methods may include physical or chemical modifications, which may alter the structure of the original drug such as in the case of cefuroxime axetil. Another method is to combine the active pharmaceutical ingredient (API) non-covalently with other compounds or ions, as is the case with cefuroxime sodium [10].

The use of APIs as cations or anions in combination with other organic biocompatible counter ions to form organic salts or ionic liquids (API-OSILs) has been used in academia and the pharmaceutical industry as an alternative formulation that expands on the properties of their inorganic counterparts [11,12,13,14,15,16,17,18]. This generation of ionic liquids may provide distinctive physicochemical properties over their original drugs and respective inorganic salts, such as melting point, cytotoxicity, or potency, making it available to new delivery systems or broadening their applications [15,16,17,18,19,20,21,22,23,24,25,26,27,28,29,30,31,32,33]. Several API-OSILs have been developed by our research team that improve the activity and pharmacokinetic parameters, such as the solubility and permeability of antibiotics (ampicillin [24,25,27], penicillin [28], amoxicillin [28], and fluoroquinolone [16,24,29]) and antivirals like hydroxychloroquine [34], as well as other APIs such as ibuprofen [23,24], naproxen [24], bone antiresorptive agents zoledronic [30], alendronic [31] and etidronic acids [35].

Organic salts of antibiotics, such as the β-lactam penicillin, have improved solubility compared to the base or sodium salt forms of ampicillin. While most salts have lower or similar water solubility to an ampicillin solution, they are orders of magnitude more soluble than base ampicillin, and most organic salts had a higher octanol partition than either base or sodium salt ampicillin [26]. Activity was also significantly affected. Notably, some of the ampicillin organic salts could reverse antibiotic resistance in *E. coli* [25]. Studies on antivirals such as hydroxychloroquine have demonstrated that incorporating these molecules with organic counter-ions can decreased the virus-induced cytopathic effect by two-fold in comparison with the original drug [34]. Furthermore, the added packing frustration provided by organic counter ions may lower the melting point of an API so as to attain a liquid state at room temperature, at which point any polymorphic tendency of an API can be considerably reduced or even eliminated, thus potentially tackling formulation and dosage issues [29,30,36].

This report presents the synthesis and characterization of five novel organic salts or ionic liquids of cefuroxime (CFX-OSILs). The anion is combined with the cations hexylpyidinium, decylpyridinium, cetylpyridinium and decanediylbispyridinium as well as cetylmethylimidazolium. The nuclear magnetic resonance (^1^H and ^13^C NMR), Fourier transform infrared (FTIR) spectroscopy and electrospray ionization tandem mass (ESI-MS) spectroscopic and elemental analysis techniques were used to ensure the unequivocal structural characterization, stoichiometry, and purity of the newly synthesized cefuroxime salts. The bioavailability of the prepared CFX-OSILs was evaluated experimentally via water solubility at 37 °C and octanol/water partition studies. Additionally, the cytotoxicity of the CFX-OSILs was tested on 3T3 murine fibroblasts, and their antimicrobial activity was tested against Gram-positive (*B. subtilis*) and Gram-negative (*E. coli* and *P. aeruginosa*) bacteria.

## 2. Materials and Methods

Commercially available reagents from Sigma-Aldrich (St. Louis, MO, USA), Alfa Aesar (Karlsruhe, Germany) and TCI Europe (Zwijndrecht, Belgium) were purchased and used as received. Standard solvents were also purchased from Laborspirit (Lisbon, Portugal) and used without further purification. The basic anion-exchange resin Amberlite 26-OH (ion-exchange capacity 0.8 eq·mL^−1^) was purchased from Supelco (St. Louis, MO, USA). ^1^H and ^13^C NMR (in APT mode) spectra in CD_3_OD (from Euriso-Top, St. Aubin, France) were recorded on Bruker AMX400 and AMX500 spectrometers at 25 °C. To perform NMR, 5 mm borosilicate tubes were used, and the sample concentration was, approximately, 20 mg/mL for ^1^H NMR and 40 mg/mL for ^13^C NMR. Chemical shifts are reported downfield in parts per million (ppm). FTIR spectra were measured on a Perkin Elmer 683 in ATR mode. The elemental analysis experiments were performed in a CHNS Series Thermo Finnigan-CE Instruments Flash EA 1112 under standard conditions (T combustion reactor 900 °C, T GC column furnace 65 °C, multiseparation SS GC column, He_2_ flow 130 mL/min and O_2_ flow 250 mL/min) at the Analysis Laboratory LAQV REQUIMTE—Chemistry Department, FCT NOVA, Portugal.

### 2.1. General Synthesis

One molar equivalent of each cation halide salt (0.5 equivalents in the case of decylbispyridinium dibromide [PyrC_10_Pyr]Br_2_) was dissolved in 1–2 mL of methanol and left to stir with 1 mL of anionic resin, Amberlyst A-26, in hydroxide form (A-26 (OH)). After 1 h, the solution was filtered, and the resin was washed thoroughly thrice with methanol. Meanwhile, cefuroxime (51 mg, 118 µmol) was dispersed in 2 mL of distilled water, to which 75 mg of ammonium bicarbonate was added under stirring. After complete dissolution, the solutions of the cations were added dropwise to the cefuroxime-buffered solution while stirring at 0 °C. The resulting solution was stirred for 15 min before being evaporated in a rotary evaporator at 40 °C and subsequently dried under high vacuum for 24 h.

### 2.2. Cytotoxicity of CFX-OSILs

Cytotoxic activities were evaluated on 3T3 cells previously acquired from Leibniz Institute DSMZ, Germany biobank (ACC173). Cells were cultured in Dulbecco’s Modified Eagle’s medium: Nutrient Mix F-12 (DMEM/F-12) (Merck, Darmstadt, Germany) supplemented with 10% fetal bovine serum (Gibco, Grand Island, NY, USA), 100 IU/mL penicillin and 100 μg/mL streptomycin (Sigma, St. Louis, MO, USA). For subculture, 3T3 cells were dissociated using trypsin-EDTA (Sigma, USA) with a split 1:5 ratio and seeded into Petri dishes with 25 cm^2^ of growth area. Culture medium was replaced every 2 days until cells reached total confluence after 3–5 days of initial seeding. Cells were maintained in a humidified atmosphere with 5% CO_2_ at 37 °C. The effects on 3T3 cells’ viability were evaluated after cells reached total confluence. Cells were treated with samples for 24 h at 100 µM. The IC_50_ value was determined (1–100 µM) for the compounds that reduced the cell viability by more than 50%. The effects were estimated via MTT (Sigma, St. Louis, MO, USA) assays as a cellular metabolic activity colorimetric analysis. Results were expressed in percentage of control (%).

### 2.3. Antimicrobial Activities

The antimicrobial activity of compounds was evaluated against *Bacillus subtilis* (ATCC 6633), *Pseudomonas aeruginosa* (ATCC 27853) and *Escherichia coli* (ATCC 25922) grown in Lysogeny broth (Sigma, Darmstadt, Germany). The microorganism growth inhibition was accompanied by the reading of optical density at 600 nm (Synergy H1 Multi-Mode Microplate Reader, BioTek^®^ Instruments, (Winooski, VT, USA). The IC_50_ was determined (0.03–100 µM) for the most active compounds (microorganism growth inhibition >50% at 100 µM). Results were expressed in percentage of growth inhibition relative to the control (growth medium with microorganism).

### 2.4. Data and Statistical Analysis

Results are presented as mean ± standard error of the mean (SEM). One-way analysis of variance (ANOVA) with Dunnett’s multiple comparison of group means to determine significant differences relative to the control treatment was accomplished (as indicated in the following manuscript https://doi.org/10.1002/9781118445112.stat06938, accessed on June 2024). Differences were considered significant at a level of 0.05 (*p* < 0.05). The IC_50_ value was estimated via the analysis of non-linear regression using the following equation:$$Y=\frac{100}{1+10\left(X-\log\left(IC50\right)\right)}$$

Calculations were performed using GraphPad v8.0.2 (GraphPad Software, La Jolla, CA, USA) software.

## 3. Results and Discussion

### 3.1. Synthesis and Characterization

Cefuroxime (CFX) was combined as an anion with five different organic cations, namely hexyl-([C_6_Py]), decyl-([C_10_Py]) and cetylpyridinium ([C_16_Py]), as well as cetylmethylimidazolium ([C_16_MIM]) and decylbispyridinium ([PyC_10_Py]) (see Figure 1).

The rationale behind the selection of these cations was to study the influence of the chain length ([C_6_Py], [C_10_Py], [C_16_Py]), the cation moiety ([C_16_Py], [C_16_MIM]) and charge ([C_10_Py], [PyC_10_Py]) on water solubility, partition coefficient, toxicity and antimicrobial activity against reference Gram-negative and Gram-positive strains.

According to our experience in the synthesis of ampicillin-based OSILs [25], a 1 M ammonium hydroxide buffer solution (pH 11.6) must be used in order to avoid the opening of the alkaline-labile β-lactam ring via exposure of the drug to the highly basic cation hydroxides (pH 14). However, in the present work, the use of this buffer led to ring opening and degradation of CFX. During the synthesis of amoxicillin-based OSILs [28], it was observed that ammonia-hydrolysate amoxicillin OSIL derivatives (seco-AMX-OSILs) were consistently obtained, despite the investigation of other methodologies. Thus, a new synthetic methodology was developed to produce the desired CFX-OSILs. Other inorganic ammonium salts were screened as potential buffers in the reaction media, which could be easily eliminated after the reaction. With these requirements in mind, ammonium carbonate and ammonium bicarbonate seemed to be worthy of investigation, as both of them decompose into ammonia, water and carbon dioxide when slightly heated.

Hence, on a first approach, we prepared 1 M solutions of each buffer at pH values of 8.4 and 7.8, respectively. We then proceeded to synthesize [NH_4_][CFX]. CFX was dissolved in the basic aqueous media to produce [NH_4_][CFX]. The reaction was allowed to continue for 10 additional minutes, after which the solvent was evaporated under reduced pressure at 40 °C. Liberation of bubbles, consistent with the formation of CO_2_ and NH_3_, was observed in both cases; however, it was observed to a greater extent in the NH_4_HCO_3_-buffered reaction. Spectroscopic characterization via ^1^H NMR revealed decomposition of CFX in both cases, but to a lesser extent with NH_4_HCO_3_ than with (NH_4_)_2_CO_3_. Moreover, the elemental analyses showed higher-than-expected nitrogen content, which is characteristic of the formation of ammonia hydrolysates via β-lactam ring opening, as well as the presence of undecomposed ammonium salts.

As ammonium bicarbonate seemed to be more promising than ammonium carbonate, due to its lower pH and faster decomposition at a lower temperature (42 °C), we optimized the reaction conditions using this base as a buffer. Our efforts focused on finding the minimum concentration of NH_4_HCO_3_ required to produce the desired CFX-OSILs without any traces of the base. Concentrations ranging from 0.1 to 0.75 M were investigated. The minimum concentration that dissolved CFX within 10 min at room temperature while rendering pure final products was found to be 0.5 M NH_4_HCO_3_, which is equivalent to four-times the amount of CFX. Lower NH_4_HCO_3_ concentrations did not dissolve CFX in the course of 1 h, and higher ones decomposed CFX. The optimization steps were performed for the synthesis of [C_16_Py][CFX]. The structure was characterized via mass spectrometry (ESI-MS), NMR (^1^H and ^13^C) and FTIR spectroscopies as well as elemental analyses and melting point determination.

The ESI-MS spectra (FIA injection) in negative mode showed the base peak to be the [M-1] at *m*/*z* 423.0 (Appendix A), consistent with the cefuroxime anion bearing the intact β-lactam moiety. On the other hand, the positive mode exhibited [M+1] at *m*/*z* 304.2 as the base peak (Appendix A), which can be assigned to the cetylpyridinium cation. The ^1^H NMR spectrum (Appendix A) showed that [C_16_Py][CFX] contained the cation and the anion in precise stoichiometric amounts. Additionally, the ^13^C NMR spectrum (Appendix A) confirmed that the compound was highly pure, as intended. The presence of the β-lactam ring was further confirmed in the FTIR spectrum (Appendix A), which contained the C=O stretching vibration at 1771 cm^−1^ (1762 cm^−1^ in original CFX)—see Figure 2 for a zoomed in view of the 1800–1500 cm^−1^ region. This spectrum also confirmed the ionization of CFX by presenting the characteristic signal of a carboxylate C=O stretching vibration at 1599 cm^−1^. The FTIR spectrum of the original CFX shows a superimposition of the analogous vibration of the corresponding carboxylic acid group with the signal at 1730 cm^−1^ from the carbamate group, making it impossible to assign it unequivocally. The amide group is responsible for the remaining C=O stretching mode at 1668 cm^−1^ (1658 cm^−1^ in original CFX). The presence of [C_16_Py][CFX] as a hydrate was confirmed via the elemental analyses.

The remaining CFX-OSILs were subsequently prepared using the same method in quantitative yields. The salts were characterized via NMR (^1^H and ^13^C) and FTIR spectroscopies, melting point and elemental analyses (see Table 1).

For all products, ^1^H NMR analysis showed that the proportion between the cation and CFX was achieved as expected. In particular, 1.0:1.0 was obtained for every CFX-OSIL with the exception of [PyC_10_Py][CFX]_2_, which yielded 1.0:2.0.

The FTIR spectra analysis of the prepared cefuroxime OSILs confirmed that ionization of cefuroxime occurred through the presence of the carboxylate’s C=O stretching band at 1598 ± 2 cm^−1^. The remaining signals in the FTIR spectra of CFX-OSILs are consistent with the structure of CFX and corresponding cations.

### 3.2. Water Solubility and Octanol-Water Partition Studies (Kow)

The solubility of the prepared cefuroxime OSILs was determined by adding known volumes of water consecutively to a certain amount of CFX-OSILs at 37 °C until full dissolution was observed. Figure 3 presents the maximum water solubility values of the prepared CFX-OSILs.

As expected, the ionization of cefuroxime into organic salts resulted in a significant increase in its water solubility, ranging from an approximately 8-to-215 times increase from that of free CFX. The increase in alkyl chain length led to a clear decrease in water solubility, as expected, with [C_6_Py][CFX] presenting 155 mg/mL while [C_10_Py][CFX] and [C_16_Py][CFX] showed similar behaviors, nonetheless higher (>7.7-fold) than the value for the original CFX. Changing the cationic group linked to the cetyl group from pyridinium ([C_16_Py]) to methylimidazolium ([C_16_MIM]) resulted in an increase in solubility from 5.5 to 11 mg/mL. Finally, the increased charge density found in the dicationic [PyC_10_Py][CFX]_2_ led to a higher hydrophilic character in comparison with the monocationic ones, which rendered the highest water solubility recorded (140 mg/mL).

Out of the five prepared CFX-OSILs, two exhibited a significantly higher partition from the aqueous phase to the organic phase (Figure 4). This was observed in the combinations of CFX with the most lipophilic cations [C_16_Py][CFX] and [C_16_MIM][CFX], as expected, due to their low solubility in water. However, the former presented a much higher value than the latter, in particular, 2.5 versus 1.7, which correlates well with the observed water solubility (Figure 3).

The CFX-OSILs based on the cation bearing the smallest alkyl chain ([C_6_Py][CFX]) and the dicationic one ([PyC_10_Py][CFX]_2_), presented water-octanol partition (Kow) values similar to the parent drug, although they are much more water-soluble. Finally, the compound with the intermediate chain size [C_10_Py][CFX] showed a slight increase in the Kow value, approximately four-times greater, in comparison with the original drug, at the same time that it is also more water-soluble.

Hence, these data show that it may be possible to modify the drug’s bioavailability by combining it with cations that have distinct lipophilic properties.

### 3.3. Biocompatibility Assessment

The biocompatibility of the synthesized CFX-OSILs and the starting materials, namely the original CFX and the cation halide salts, was studied in 3T3 murine fibroblasts using the MTT colorimetric assay. Figure 5 plots the percentage of viable cells after 24 h exposure to the CFX-OSILs at 100 µM (the maximum dose tested in the antimicrobial studies).

At a concentration of 100 µM, the three CFX-OSILs bearing the cations with the longest alkyl chains ([C_10_Py][CFX], [C_16_Py][CFX] and [C_16_MIM][CFX]) were found to be more toxic to 3T3 murine fibroblasts than the original drug. This was also observed for the corresponding cation halide salts. In turn, [C_6_Py][CFX] and [PyC_10_Py][CFX]_2_ showed no additional detrimental effect on the fibroblasts in comparison with CFX. This is consistent with the good biocompatibility shown by the corresponding cation halide salts. The IC_50_ values were determined via tracing dose–response curves (in ESI). The results showed that [C_16_Py]Cl was approximately two-times less toxic than its methylimidazolium sibling (Table 2). However, the combination of CFX with these compounds results in the formation of similarly toxic CFX-OSILs, with IC_50_ values of 11.68 and 13.94 µM, respectively. Finally, [C_10_Py][CFX] is approximately five-times less toxic than these CFX-OSILs, as expected due to its smaller alkyl chain.

### 3.4. Evaluation of the Antimicrobial Activity

The developed CFX-OSILs, as well as CFX and the corresponding cation halide salts, were incubated with Escherichia coli (ATCC 25922), Pseudomonas aeruginosa (ATCC 27853) and Bacillus subtilis (ATCC 6633) for 6 h in concentrations between 0.03 and 100 µM. Table 3 presents the calculated IC_50_ values.

All of the analyzed CFX-OSILs, except for the dicationic compound [PyC_10_Py][CFX]_2_, presented reduced antimicrobial activity against the studied bacterial strains. Previous works [16,24,25,26,27,28,29] suggested that the most lipophilic compounds would have enhanced antimicrobial activity. However, this was not the case. This result is not completely surprising because cefuroxime is already one of the most potent antibiotics ever developed. Nonetheless, [PyC_10_Py][CFX]_2_ exhibited comparable activity to CFX against *P. aeruginosa* and *B. subtilis*, and was even 1.3-times more effective against *E. coli.* The corresponding cation bromide salt did not show activity against these strains and is also non-toxic to normal cells. Moreover, the CFX-OSIL displays over 215-times greater solubility in water than CFX and has a similar lipophilic character. Therefore, it could serve as a starting point for developing a novel formulation of this antibiotic with enhanced bioavailability and excellent antimicrobial activity.

## Data Availability

Data are available in a publicly accessible repository.

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
