# Peer review of "Synthesis, Characterization, Bioavailability and Antimicrobial Studies of Cefuroxime-Based Organic Salts and Ionic Liquids"

_pharmaceutics, 2024, doi:10.3390/pharmaceutics16101291_

Round 1
Reviewer 1 Report
Comments and Suggestions for Authors
This paper presents important information on how to make an antibiotic more effective. It is marked by some careless writing errors.
Here are some errors:
line 103 Honeywell and standard solvents were also purchased from Laborspirirt and used without further purification.
What does the word Honeywell refer to? The vendor name Laborspirirt is misspelled. Other vendors get the city, province, and country listed, why not this one? Why not be consistent?
Line 117 [PyC10Py]Br2) This, at this point in the paper, needs explanation and citation.
Line 127 DSMZ biobank needs a citation, geographic location and/or web link
MTT on line 137 needs explanation
Line 142. Lysogeny broth (Darmstadt, Germany). The name of the company that produced the broth is missing. Does that company produce only one kind of lysogeny broth?
Line 150 One-way analysis of variance (ANOVA) with Dunnett’s multiple comparison of group needs a citation
Line 175 amoxicillin-based OSILs [], the reference number in the [ ] is missing
Line 172-3 opening of the alkaline-labile β-lactam ring by exposure of the drug to the highly basic cation hydroxides (pH 14) ref.
the reference number is missing
The abbreviation Kow is first used on line 263, but is not explained until line 269 that it is explained as the octanol-water partition coefficient values (Kow,
Line 283 were found to be more toxic More toxic to what? I assume to 3T3 murine fibroblasts, but that should be said.
Author Response
Reviewer 1 (R1) considered that “This paper presents important information on how to make an antibiotic more effective. It is marked by some careless writing errors”.
Authors: Thank you for your comments. The writing errors will be corrected.
R1: line 103 Honeywell and standard solvents were also purchased from Laborspirirt and used without further purification. What does the word Honeywell refer to? The vendor name Laborspirirt is misspelled. Other vendors get the city, province, and country listed, why not this one? Why not be consistent?
Authors: Thank you for the comment. The required information was added.
R1: Line 117 [PyC10Py]Br2) This, at this point in the paper, needs explanation and citation.
Authors: The chemical name of the compound (decyl-bis(pyridinium) dibromide) before abbreviation was added.
R1: Line 127 DSMZ biobank needs a citation, geographic location and/or web link
Authors: The required information was added.
R1: MTT on line 137 needs explanation
Authors: The explanation about MTT (colorimetric assay in order to evaluate cell metabolic activity) was added.
R1: Line 142. Lysogeny broth (Darmstadt, Germany). The name of the company that produced the broth is missing. Does that company produce only one kind of lysogeny broth?
Authors: The missing information was added. This company produce only one kind of lysogeny broth.
R1: Line 150 One-way analysis of variance (ANOVA) with Dunnett’s multiple comparison of group needs a citation
Authors: The citation about ANOVA with Dunnett´s multiple comparison group was added.
R1: Line 175 amoxicillin-based OSILs [], the reference number in the [ ] is missing
Authors: The reference number was added.
R1: Line 172-3 opening of the alkaline-labile β-lactam ring by exposure of the drug to the highly basic cation hydroxides (pH 14) ref. the reference number is missing
Authors: The reference number was added.
R1: The abbreviation Kow is first used on line 263, but is not explained until line 269 that it is explained as the octanol-water partition coefficient values (Kow).
Authors: The abbreviation Kow as octanol-water partition coefficient is now explained on line 263 as suggested.
R1: Line 283 were found to be more toxic More toxic to what? I assume to 3T3 murine fibroblasts, but that should be said.
Authors: As suggested, we complete the sentence: “more toxic than 3T3 murine fibroblasts”.
Reviewer 2 Report
Comments and Suggestions for Authors
The manuscript ID pharmaceutics-3124703 entitled “Synthesis, Characterization, Bioavailability and Antimicrobial Studies of Cefuroxime-Based Organic Salts and Ionic Liquids” by Faísca et al. can be accepted to publication in Pharmaceutics after minor revision.
Specific comments
According to the Authors, a synthesis and characterization of five novel organic salts or ionic liquids of cefuroxime (CFX-OSILs) by NMR, ESI-MS, FTIR and elemental analysis techniques was performed. Next, the bioavailability of the prepared CFX-OSILs was evaluated experimentally by water solubility at 37 °C and octanol/water partition studies. Moreover, the cytotoxicity of the CFX-OSILs was tested on 3T3 murine fibroblasts, and their antimicrobial activity was tested against Gram-positive (B. subtilis) and Gram-negative (E. coli and P. aeruginosa) bacteria. The study confirmed that water solubility of cefuroxime organic salts was from 8 to 200 times greater than the original drug at 37 °C. Additionally, the most soluble compounds have a very low octanol-water partition, similar to cefuroxime, while more lipophilic salts partition predominantly to the organic phase. The obtained data also showed that only the cefuroxime salts combined with longer alkyl chain cations possess higher cytotoxicity than the original drug whereas most salts lost in vitro antibacterial activity against the studied bacterial strains. Only [PyC10Py][CFX]2 retained the activity against the studied bacterial strains (comparable activity to CFX against P. aeruginosa and B. subtilis, and 1.3 times more effective against E. coli). Moreover, the dicationic structure of [PyC10Py][CFX]2 offered the highest water solubility (140 mg/mL). Additionally, the corresponding cation bromide salt did not show activity against these strains, and was also non-toxic to normal cells. Therefore, the authors conclude that it could serve as a starting point for developing of a novel formulation of this antibiotic with enhanced bioavailability and higher antimicrobial activity. The experiments had been correctly planned and performed as well as the manuscript was well written. The references are also adequate to the presented study. The obtained results should be interesting for the readers, so the manuscript can be published in Pharmaceutics after minor correction:
1) How many replications were made during the experiments described in the manuscript?
2) Editorial mistakes:
Page 2, line 76; β -lactam
Page 3, line 117; [PyC10Py]Br2)
Page 5, line 175; OSILs [],
Page 5, line 201; NH4HCO3
Page 10, line 313; theCFX-OSIL
Author Response
Reviewer 2 (R2) considered that “The manuscript ID pharmaceutics-3124703 entitled “Synthesis, Characterization, Bioavailability and Antimicrobial Studies of Cefuroxime-Based Organic Salts and Ionic Liquids” by Faísca et al. can be accepted to publication in Pharmaceutics after minor revision.”
Authors: Thank you your comments.
R2: Specific comments: “According to the Authors, a synthesis and characterization of five novel organic salts or ionic liquids of cefuroxime (CFX-OSILs) by NMR, ESI-MS, FTIR and elemental analysis techniques was performed. Next, the bioavailability of the prepared CFX-OSILs was evaluated experimentally by water solubility at 37 °C and octanol/water partition studies. Moreover, the cytotoxicity of the CFX-OSILs was tested on 3T3 murine fibroblasts, and their antimicrobial activity was tested against Gram-positive (B. subtilis) and Gram-negative (E. coli and P. aeruginosa) bacteria. The study confirmed that water solubility of cefuroxime organic salts was from 8 to 200 times greater than the original drug at 37 °C. Additionally, the most soluble compounds have a very low octanol-water partition, similar to cefuroxime, while more lipophilic salts partition predominantly to the organic phase. The obtained data also showed that only the cefuroxime salts combined with longer alkyl chain cations possess higher cytotoxicity than the original drug whereas most salts lost in vitro antibacterial activity against the studied bacterial strains. Only [PyC10Py][CFX]2 retained the activity against the studied bacterial strains (comparable activity to CFX against P. aeruginosa and B. subtilis, and 1.3 times more effective against E. coli). Moreover, the dicationic structure of [PyC10Py][CFX]2 offered the highest water solubility (140 mg/mL). Additionally, the corresponding cation bromide salt did not show activity against these strains, and was also non-toxic to normal cells. Therefore, the authors conclude that it could serve as a starting point for developing of a novel formulation of this antibiotic with enhanced bioavailability and higher antimicrobial activity. The experiments had been correctly planned and performed as well as the manuscript was well written. The references are also adequate to the presented study. The obtained results should be interesting for the readers, so the manuscript can be published in Pharmaceutics after minor correction”
Authors: Thank you for your detailed comments.
R2: How many replications were made during the experiments described in the manuscript?
Authors: The experiments presented in the manuscript were performed at least 3 times (3 replications).
R2: Editorial mistakes:
Page 2, line 76; β -lactam
Page 3, line 117; [PyC10Py]Br2)
Page 5, line 175; OSILs [],
Page 5, line 201; NH4HCO3
Page 10, line 313; theCFX-OSIL
Authors: All editorial mistakes were corrected.
Reviewer 3 Report
Comments and Suggestions for Authors
The manuscript by Faísca et al. presents the derivatization of Cefuroxime into organic salts to improve its water solubility. The authors used different cations such as imidazolium and pyridinium derivatives and combined it with Cefuroxime to produce salts. The study is well executed and the methods used are correctly presented. I recommend publication of the manuscript as it is (minor language corrections should be made at the proofreading stage).
I’m only wondering why the authors have chosen the most common imidazolium and pyridinium cations, known for their toxicity (additionally derived from fossil fuels) instead of more biocompatibile, bio-derived cations as the main goal of the study was only derivatization of Cefuroxime into a salt. The authors could add some comment on the toxicity of these cations (in the study only the effect on fibroblasts is presented).
Comments on the Quality of English LanguageMinor English corrections should be made.
Author Response
Reviewer 3 (R3) considered that e “The manuscript by Faísca et al. presents the derivatization of Cefuroxime into organic salts to improve its water solubility. The authors used different cations such as imidazolium and pyridinium derivatives and combined it with Cefuroxime to produce salts. The study is well executed and the methods used are correctly presented. I recommend publication of the manuscript as it is (minor language corrections should be made at the proofreading stage).”
R3: I’m only wondering why the authors have chosen the most common imidazolium and pyridinium cations, known for their toxicity (additionally derived from fossil fuels) instead of more biocompatibile, bio-derived cations as the main goal of the study was only derivatization of Cefuroxime into a salt.
Authors: According to our previous experience, the presence of imidazolium and pyridinium cations combined with antibiotic anions (e.g. beta-lactams or fluoroquinoline) improved significantly the antimicrobial activity comparing to other classes of cations such as tetra-alkylammonium or tetra-alkylphosphonium. In this context, we decided to select imidazolium or pyridinium cations modulating the size of the alkyl chain [CnMIM] or [CnPyr].
R3: The authors could add some comment on the toxicity of these cations (in the study only the effect on fibroblasts is presented).
Authors: The toxicity of the selected organic cations were checked using fibroblasts as already reported in previous works. In the literature, fibroblasts were also tested as healthy cell lines. Additional comment about toxicity of the selected cations was included in the manuscript.